# From Tick to Test: A Comprehensive Review of Tick-Borne Disease Diagnostics and Surveillance Methods in the United States

**DOI:** 10.3390/life13102048

**Published:** 2023-10-13

**Authors:** Sean Rowan, Nazleen Mohseni, Mariann Chang, Hannah Burger, Mykah Peters, Sheema Mir

**Affiliations:** College of Veterinary Medicine, Western University of Health Sciences, Pomona, CA 91766, USA; sean.rowan@westernu.edu (S.R.);

**Keywords:** tick-borne diseases, TBD, surveillance, ticks, molecular diagnostics, One Health

## Abstract

Tick-borne diseases (TBDs) have become a significant public health concern in the United States over the past few decades. The increasing incidence and geographical spread of these diseases have prompted the implementation of robust surveillance systems to monitor their prevalence, distribution, and impact on human health. This comprehensive review describes key disease features with the geographical distribution of all known tick-borne pathogens in the United States, along with examining disease surveillance efforts, focusing on strategies, challenges, and advancements. Surveillance methods include passive and active surveillance, laboratory-based surveillance, sentinel surveillance, and a One Health approach. Key surveillance systems, such as the National Notifiable Diseases Surveillance System (NNDSS), TickNET, and the Tick-Borne Disease Laboratory Network (TBDLN), are discussed. Data collection and reporting challenges, such as underreporting and misdiagnosis, are highlighted. The review addresses challenges, including lack of standardization, surveillance in non-human hosts, and data integration. Innovations encompass molecular techniques, syndromic surveillance, and tick surveillance programs. Implications for public health cover prevention strategies, early detection, treatment, and public education. Future directions emphasize enhanced surveillance networks, integrated vector management, research priorities, and policy implications. This review enhances understanding of TBD surveillance, aiding in informed decision-making for effective disease prevention and control. By understanding the current surveillance landscape, public health officials, researchers, and policymakers can make informed decisions to mitigate the burden of (TBDs).

## 1. Introduction

Tick-borne diseases (TBDs) have emerged as a growing public health concern in the United States, posing significant challenges to human health and necessitating effective surveillance strategies for prevention and control [1]. The incidence and geographic distribution of TBDs have been steadily increasing over the past few decades, leading to substantial morbidity and occasionally mortality in affected individuals [2]. Understanding the rise of these diseases, their impact on public health, and the critical role of surveillance is paramount in developing effective prevention and control measures.

Ticks are vectors for a wide range of pathogens, including bacteria, viruses, and parasites, which can cause debilitating illnesses in humans [3]. Some of the most common tick species in the U.S., known for transmitting diseases, include: Blacklegged Tick (*Ixodes scapularis*), Western Blacklegged Tick (*Ixodes pacificus*), American Dog Tick (*Dermacentor variabilis*), Brown Dog Tick (*Rhipicephalus sanguineus*), Lone Star Tick (*Amblyomma americanum*), Gulf Coast Tick (*Amblyomma maculatum*), Rocky Mountain Wood Tick (*Dermacentor andersoni*), Pacific Coast Tick (*Dermacentor occidentalis*), and Soft Tick (*Ornithodoros* spp.) [4]. The most recognized tick-borne disease in the United States is Lyme disease, caused by the bacterium *Borrelia burgdorferi*, but numerous other TBDs, such as anaplasmosis, babesiosis, Powassan virus disease, ehrlichiosis, and Rocky Mountain spotted fever, have also been identified and pose unique challenges to public health surveillance [5].The impact of TBDs on public health is significant. Lyme disease alone accounts for thousands of cases reported annually in the United States, with symptoms ranging from mild flu-like illness to severe complications affecting multiple body systems, including the nervous system, joints, and heart [6]. Other TBDs can cause similar or distinct clinical presentations, with varying levels of severity. Furthermore, the physical, emotional, and economic burden imposed by these diseases on individuals and communities cannot be overlooked [7].

Effective surveillance plays a crucial role in the prevention and control of TBDs. Surveillance systems enable the systematic collection, analysis, and interpretation of data on disease incidence, geographic distribution, temporal trends, and risk factors [7]. Such information is vital for identifying high-risk areas, detecting disease outbreaks, assessing the effectiveness of control measures, and guiding public health interventions [8]. Timely and accurate surveillance data allow public health officials and policymakers to allocate resources efficiently, develop targeted prevention strategies, and inform healthcare providers and the public about the risks associated with TBDs [7,8].

This review article comprehensively examines the current landscape of tick-borne disease surveillance in the United States. By analyzing the strategies, challenges, and advancements in surveillance methods, this article provides valuable insights into the prevention, detection, and management of TBDs. The findings highlight the importance of a multidisciplinary approach, emphasizing the need for collaboration among public health agencies, researchers, healthcare providers, and the community. With continued efforts to improve surveillance systems, the United States can effectively combat the growing threat of TBDs and protect the health of its population.

## 2. Tick-Borne Diseases in the United States

### 2.1. Lyme Disease

Lyme disease is the most common TBD in the United States [9]. The bacteria *Borrelia burgdorferi* is the etiological agent of the infection and is transmitted to humans and animals through *Ixodes* ticks. There are approximately 476,000 diagnosed cases in the United States per year with *Ixodes scapularis* being the main vector of the infection [10]. Most cases of Lyme disease within the United States of America (USA) are in the mid-Atlantic, Northeast, and Upper Midwest regions [11]. Figure 1 gives the annual reported cases of Lyme disease for the year 2021 in the USA, according to the Centers for Disease Control and Prevention (CDC) (data from some reporting areas may be incomplete due to the 2019 coronavirus disease (COVID-19) pandemic). The most common clinical manifestation of Lyme disease is erythema migrans, a rash on the skin about 10 cm in size, which occurs in approximately 80% of patients affected by Lyme disease. Other clinical manifestations that occur weeks to months after the initial tick transmission include Lyme arthritis, neuroborreliosis, and borrelial lymphocytoma [12]. Another disease agent that is closely related to the bacteria causing Lyme disease is *Borrelia mayonii*, transmitted to humans by the blacklegged (deer) tick. This was first identified in Minnesota in 2013 [13].

### 2.2. Anaplasmosis

*Anaplasma* (*A*) *phagocytophilum* is the etiological agent for human granulocytic anaplasmosis (HGA) [15]. *A. phagocytophilum* is a bacteria transmitted by the blacklegged tick, *Ixodes scapularis* [6]. *A. phagocytophilum* primarily targets neutrophils and is geographically prominent in the northeastern and north central states as well as northern California [16]. With approximately 4151 cases reported per year, there is still a steady increase in cases within these regions [17]. The geographic distribution for HGA is very similar to Lyme disease because both diseases share the same primary hosts and primary vectors. The six states with the highest incidence for HGA are Rhode Island, Minnesota, Connecticut, Wisconsin, New York, and Maryland [18]. In New York, anaplasmosis has become the second most common TBD [19]. With the continuing expansion of the blacklegged tick in non-endemic regions of the USA, *A. phagocytophilum*, and other pathogens that this tick carries, will eventually become a greater burden to human health. The blacklegged tick can carry and transmit a variety of different pathogens and with an increase in this tick in endemic areas, patients may carry co-infections and experience more severe illnesses that require more extensive treatments [20]. A lack of awareness and under reporting of these TBDs, like anaplasmosis, can lead to undiagnosed and untreated cases in both endemic and novel regions of the disease [19]. Clinical manifestations of anaplasmosis include fever, headache, leukopenia, thrombocytopenia, and myalgias. Rashes are uncommon in HGA and occur in less than 10% of patients. Neurological manifestations are also uncommon in HGA [16].

### 2.3. Babesiosis

*Babesia microti*, an intraerythrocytic protozoan, is the main etiological agent of human tick-borne babesiosis [21]. Infection by the *Babesia* species occurs primarily through *Ixodes* ticks and the primary reservoir species in the USA is the white-footed mouse [22]. The *Babesia* species is endemic to the northeastern and upper Midwestern states, but its geographical range has expanded past these endemic regions and can range from Maine to Maryland. Most cases of transmission occur in the summer due to the increased number of ticks during this time [23]. The number of reported cases of babesiosis with *Babesia microti* is approximately 2000 cases per year [17]. Ticks are infected with the *Babesia* species by ingesting host erythrocytes that contain the pathogen. The parasite reaches the tick’s salivary glands and is transmitted transstadially from one tick stage to the next or transovarially to the eggs [24].Clinical signs of babesiosis include fever, chills, sweats, anorexia, headache, myalgia, nausea, and arthralgia. Laboratory indicators, which can include low hemoglobin and low hematocrit levels, reflect hemolytic anemia due to the *Babesia* species’ invasion of red blood cells. Immunocompromised and elderly patients may experience more severe infections which include respiratory distress, pulmonary edema, congestive heart failure, renal failure, or splenic rupture. Co-infections of *Babesia* with other tick-borne infections may increase the severity of acute symptoms [23].

### 2.4. Powassan Virus Disease

Powassan virus disease was initially discovered in a town in Ontario, Canada when it was isolated from the brain of a 5-year-old boy who died from severe encephalitis [25]. Like dengue, yellow fever, and West Nile encephalitis, Powassan virus belongs to the vector-borne flavivirus group [26]. This flavivirus is encoded by a single polypeptide that replicates within host cells [27]. The primary vectors responsible for transmitting the disease are hard-bodied *Ixodes* ticks, particularly *I. cookei* (the groundhog tick) and *I. scapularis* [26]. Unlike many flaviviruses that are transmitted by mosquitoes, there have been no reported cases of Powassan virus disease transmitted by mosquitoes, nor is there evidence supporting their competence as vectors for viral transmission. Interestingly, Powassan virus can be transmitted in as little as 15 min of tick feeding [27]. This quick transmission is attributed to the virus already being present in the salivary glands during feeding, unlike other non-viral TBDs that reside in the tick’s midgut [28]. The quick transmission along with the neurological sequelae of Powassan virus disease are devastating, with a mortality rate of 10% [27,28]. Given the severity of this disease, there is a pressing need for its detection and ongoing research in the development of diagnostic tools for global surveillance purposes.

### 2.5. Ehrlichiosis

*Ehrlichia (E) chaffeensis* and *Ehrlichia ewingii* are causative agents of human tick-borne ehrlichiosis. These agents of ehrlichiosis are intracellular Gram-negative bacteria that replicate in granulocytes of the host. *E. chaffeensis* is the etiological agent of human monocytic ehrlichiosis (HME) while *E. ewingii* is the etiological agent of human Ewingii ehrlichiosis (HEE) [16]. Infection of HME is primarily through the Lone Star Tick, *Amblyomma americanum*. *E. chaffeensis* is geographically located in the southeast, south central, and Midwest states and targets monocytes and macrophages in infected individuals [16]. *Ehrlichia chaffeensis* is reported to cause 1377 cases annually of HME [17]. Ehrlichia ewingii differs by primarily targeting neutrophils in infected patients and is most prominent in the southeast, south central, and Midwestern states. Clinical manifestations are similar to anaplasmosis, which includes fever, headache, leukopenia, thrombocytopenia, myalgias, and arthralgias. Rashes occur in approximately 10% of cases and manifest on the face, palms, and soles of the feet. Neurological manifestations are identified in about 20% of patients with HME [Ismail, 2010 #5736]. Since 2009, more than 115 instances of ehrlichiosis attributed to *E. muris eauclairensis* have also been detected in patients in the Upper Midwest. The tick species responsible for spreading this new subspecies is *Ixodes scapularis*. In terms of clinical presentation, it typically mirrors the symptoms associated with infections induced by *E. chaffeensis* and *E. ewingii* [29].

### 2.6. Spotted Fever Rickettsioses

Rocky Mountain spotted fever (RMSF) is one of the most common and severe tick-borne rickettsial infections in North America [30]. It is caused by the Gram-negative obligate intracellular bacterium *Rickettsia rickettsii*, leading to an acute febrile illness. Numerous genera and species of *ixodid* ticks are known to carry *rickettsiae* naturally. However, transmission most often occurs after a bite from ticks such as the American dog tick (*Dermacentor variabilis*), the Rocky Mountain wood tick (*Dermacentor andersoni*), or the brown dog tick (*Rhipicephalus sanguineus*) [31,32]. Other hematophagous arthropods like lice, mites, mosquitoes, and fleas can also serve as vectors for Rickettsial diseases [33,34]. RMSF is frequently misdiagnosed due to its initial presentation of nonspecific symptoms, including fever, headache, rash, myalgia, and nausea. The disease progresses rapidly and can be fatal if not detected and treated within the first five days of illness [30]. While the immuno-fluorescence assay (IFA), which compares IgG titers between acute and convalescent samples, is considered the gold standard for detection, it is important to note that most diagnostic results are not available within the first five days of illness, which coincides with the ideal treatment period. Consequently, the primary approach to diagnosis remains primarily clinical [30]. The geographical range of recognized tick-associated *rickettsiae* has expanded since the 1980s. Global surveillance reports these infections in livestock worldwide, with wildlife infections being more frequently reported in Europe and Africa. Additionally, there is a higher frequency of tick-associated *rickettsiae* infections in dogs and cats in North America. Notably, national surveillance in the United States has identified most cases in states such as Oklahoma, Arkansas, Missouri, Tennessee, and North Carolina [30]. The neglect of rickettsial infections manifests as under-recognition, leading to under-treatment due to a lack of available diagnostics, rather than the absence of an effective specific treatment [34]. This under-recognition significantly hampers the prompt administration of antibiotic treatment, resulting in mortality rates rising as high as 20–30% [32,34]. Moreover, this limitation impairs our ability to accurately assess the global burden of RMSF.

*R. parkeri* rickettsiosis, also known as American boutonneuse fever, is a tick-borne illness caused by the bacterium *Rickettsia parkeri*. It is primarily transmitted to humans through the bite of infected Amblyomma maculatum ticks, commonly referred to as Gulf Coast ticks. This disease is prevalent in certain regions of the United States, particularly in the southeastern states [35]. Pacific Coast tick fever, on the other hand, is a term that can refer to several different rickettsial diseases transmitted by ticks on the western coast of the United States. These diseases are caused by various species of Rickettsia, including *Rickettsia philipii*. The ticks responsible for transmitting these pathogens are primarily found in the western coastal states [36]. Both R. parkeri rickettsiosis and Pacific Coast tick fever share similarities in their mode of transmission through tick bites and can lead to a range of symptoms, including fever, rash, and other flu-like symptoms [36].

### 2.7. Other Tick-Borne Diseases (TBDs)

There are several lesser-known TBDs of concern in the USA, such as tularemia, Colorado tick fever, tick-borne relapsing fever, Heartland virus, and Bourbon virus. Tularemia, caused by the bacterium *Francisella tularensis*, can be transmitted through tick and insect bites, handling infected tissue, consuming undercooked meat from infected mammals, and rarely through inhalation of bacteria. It typically results in ulceroglandular disease, characterized by painful regional lymphadenopathy and a cutaneous eschar at the site of the tick bite [37].

Colorado tick fever (CTF) is a rare viral disease transmitted primarily through the bite of an infected Rocky Mountain wood tick, *Dermacentor andersoni*. These ticks are found in the western United States and Canada at elevations of 4000–10,000 feet, usually in grassy areas near sage or other brush [38]. CTF symptoms include fever, headache, myalgia, and fatigue, which can persist for several weeks. Although the reported cases of CTF in the United States are relatively low, it remains a cause of febrile illness in the Rocky Mountain regions [38]. Tick-borne relapsing fever (TBRF) is transmitted by *Ornithodoros* ticks, commonly infected with *Borrelia hermsii* and *Borrelia turicata.* Unlike other ticks found in tall brush and grass, *Ornithodoros* tick species that spread TBRF live in rodent burrows [37]. The highest concentrations of TBRF in the United States are primarily found in the western states, including the Cascade, San Bernardino, Sierra Nevada, and Rocky Mountain ranges [39]. TBRF symptoms typically appear approximately seven days after exposure and include high fever, headache, muscle and joint aches, and nausea. The fever lasts for about three days, accompanied by episodes of rigors, increased heart rate, and elevated blood pressure. Profuse sweating and a decline in fever then follows. It is crucial to consider TBRF in patients experiencing recurrent fevers, as untreated cases can lead to cardiac and neurological complications with potential long-term effects [40]. While these diseases are rare, other emerging threats in North America should also be noted. *Borrelia miyamotoi*, a tick-borne pathogen that infects ticks along with other pathogens, has recently been identified as a cause of relapsing fever. It may share a similar geographic range with *Borrelia burgdorferi*, the bacterium responsible for Lyme disease, as they both infect the same *Ixodes* ticks [26,37]. Southern tick-associated rash illness, spread by the Lone Star tick, is an emerging infection in the southeastern United States. The infectious cause of this illness is unknown, and it presents with a rash like erythema migrans [37]. 

Heartland virus and Bourbon virus are both tick-borne illnesses caused by viruses. Heartland virus was first identified in the United States in 2009, and it is primarily transmitted by the Lone Star tick. It can lead to flu-like symptoms and, in severe cases, may result in hospitalization [41]. Bourbon virus, discovered in 2014, is another tick-borne virus found in the Midwest and southern United States. It can cause fever, fatigue, rash, and other flu-like symptoms. Although it is rare, Bourbon virus can lead to severe illness and, in some cases, fatalities. Both viruses serve as a reminder of the potential health risks associated with tick bites, emphasizing the importance of preventive measures and seeking medical attention if symptoms arise after possible exposure to ticks [42]. Improvements are needed in diagnostic tools to detect and differentiate all tick-borne viral diseases, regardless of their prevalence within the region.

## 3. Surveillance Methods (Table 1)

### 3.1. Passive Surveillance

Disease surveillance is characterized as the systematic collection, analysis, and dissemination of data on infections that are significant to public or animal health [43]. This surveillance serves to inform appropriate actions to prevent or limit further disease spread. Various methods can facilitate the acquisition of information on ticks and TBDs. One such method is passive surveillance, where health jurisdictions receive reports from hospitals, clinics, public health units, or other sources [43]. This relatively inexpensive strategy covers large areas and provides critical information for monitoring communities. Data from across the literature involve both passive and inductive surveillance of TBDs across the USA [44]. Some of the surveillance in the literature, conducted within specific state regions, focuses on determining the presence of vectors and pathogens, rather than the density of tick populations or the prevalence of pathogens among those populations [44]. Passive surveillance uses classified county status and identifies the presence of ticks but has its limitations in determining the prevalence of pathogens in tick populations. These limitations occurred due to factors such as human, pet, and wildlife travel, which make the precise location of tick collection less reliable [44]. By utilizing passive surveillance systems, some have sought to close the gap between human case data and entomological surveys [45]. This study’s findings demonstrate the correlation between measures derived from passive surveillance of ticks biting humans and their pathogen testing, with the incidence of human babesiosis and anaplasmpsis. Consequently, passive surveillance holds promise in establishing a link between human behavior and the risk associated with tick-borne diseases, enabling targeted public health interventions [45].

**Table 1 life-13-02048-t001:** Summary of the different surveillance methods for tick-borne pathogens in the USA.

Surveillance Method	Description	Advantages	Disadvantages
Active Surveillance	Involves actively seeking out and testing ticks or individuals for tick-borne pathogens.	Provides real-time data on tick infection rates and pathogen presence. Allows for targeted sampling in high-risk areas or populations. Can detect emerging pathogens.	Resource-intensive and time-consuming. May not capture the full range of tick species or locations. Relies on active participation of individuals or organizations.
Passive Surveillance	Relies on reports from healthcare providers, laboratories, or the public regarding ticks and diagnosed cases of tick-borne diseases.	Relatively low cost and effort. Provides information on human cases and associated pathogens. Can cover a large geographic area. Can capture severe or unusual cases.	Underreporting and underdiagnosis may occur. May lack comprehensive data on tick species or infection rates. Dependent on the willingness and awareness of reporting entities and individuals.
Sentinel Surveillance	Selects specific sites or individuals (sentinels) to provide ongoing data on tick abundance, infection rates, and disease cases.	Provides targeted data from high-risk areas or populations. Allows for long-term monitoring and trend analysis. Can identify early warning signs of disease emergence or changes in patterns. Enables collaboration and coordination between multiple stakeholders.	Limited to selected sentinel sites or populations. May not capture all tick species or locations. Requires continuous resources and commitment. Results may not be generalizable to larger areas.

### 3.2. Active Surveillance

Active surveillance is a system characterized by employing staff members to regularly contact healthcare providers or the population to gather information about health conditions. While active surveillance generally provides the most accurate and timely information, it can also be the most expensive approach. Passive and active surveillance may be used together [46] to assess entomological measures of risk, with active surveillance being particularly useful for filling in gaps where passive submissions are lacking [45]. Active surveillance is important in describing the density and pathogen prevalence in host-seeking ticks, the main predictors of acarological risk [45]. This is particularly true with nymphs, as they are less accounted for with passive testing [45]. In one study, ticks were collected over a two-year period from March 2019 to December 2020, with sampling locations strategically chosen based on the presence of suitable tick habitats [44]. Active surveillance efforts for surveying tick populations have shown high specificity, indicating that the collection of several specimens from a site indicates a self-sustaining, reproducing tick population at that location [44]. Comparatively, host-seeking ticks provide a more precise spatial distribution than ticks collected directly from hosts. Limitations to active surveillance revolve around cost, time, and sample collection availability [47]. Nevertheless, with its high specificity, active surveillance proves to be a valuable tool for field investigations and specific community-based studies.

### 3.3. Laboratory-Based Surveillance

Laboratory-based surveillance relies on collecting information about bacteria causing diseases that have been identified through laboratory testing of ill individuals or animals. Clinical laboratories isolate and identify samples from patient specimens, which are then submitted to state public health laboratories for further characterization or reporting. Currently, laboratory-based surveillance in the USA has mainly focused on Lyme disease, as is evident from the available literature [48]. Public health surveillance for Lyme disease typically requires clinical follow-up on positive laboratory reports. Using laboratory-based surveillance solely on positive laboratory reports may serve as a potential alternative to improve standardization in already high-incidence areas. In a reported study, the number of reported cases using a laboratory-based approach to surveillance in high-incidence states, with the recommended two-tier algorithm, was on average 1.2 times higher than what is reported by the Centers for Disease Control and Prevention (CDC) [49]. Adopting this approach in high-incidence states will enhance standardization and reduce the burden on public health systems [49]. Consequently, public health resources can then focus on prevention messaging, exploration of novel prevention strategies, and alternative data sources to provide comprehensive information on the epidemiology of Lyme disease and other tick-borne pathogens [49].

### 3.4. Sentinel Surveillance

In North America, ticks are dispersed from the USA to Canada through migratory birds and once dispersed, their increased survival is due to climate change and land modifications allowing greater transmission of tick-borne pathogens throughout North America [50]. In Quebec, Canada, sentinel surveillance has captured regional trends in Lyme disease in a large geographical area [51]. Sentinel surveillance involves repeated sampling from a population and can identify annual variations in disease or species density within a population. After half a decade of surveillance, maps were generated for these geographical regions based on tracking spatial and temporal variation in Lyme disease and it was found that these maps were equivalent to complex risk assessment maps based on multiple data points. This study showed that though sentinel surveillance can be a cost-effective approach to monitor spatiotemporal trends of TBDs in large geographical areas over time it can be used to inform the public health authorities about the risk of Lyme disease to human and animal populations within a large geographical region. More sentinel studies need to be conducted throughout the USA to have a better understanding of spatial and temporal distribution of ticks and the pathogens they harbor over time [51]. 

### 3.5. One Health Approach (Table 2)

Prevention of TBDs is dependent on the surveillance and interactions between humans, animals, and the environment [43]. Understanding where, when, and how a pathogen transmits within a population can bring a better understanding of how to control disease outbreaks. This is important for tick-borne pathogens due to ticks being one of the main vectors for transmission of pathogens to humans, domestic animals, and livestock. The most common use of surveillance is to monitor ‘at risk’ populations and this is reliant on submitted patient or infected animal disease samples [43]. A study in the mid-1990s in a small community in Maryland, identified that the risk of Lyme disease was rare in this community due to over 90% of collected ticks from the environment being *A. americanum*, the lone star tick [7]. This study identified that the risk of Lyme disease in both humans and domestic animals was disproportionally lower than what was previously perceived due to the blacklegged tick being the main vector for Lyme disease. Another study conducted in Monmouth County, New Jersey, had investigated the predominant tick species submitted over time, and during 2006–2011, *I. scapularis*, the blacklegged tick, made up the majority of the submissions, followed by *A. Americanum* [52]. This continued study then identified that during 2012 to 2016, *A. americanum* replaced *I. scapularis* as the most frequently submitted species. Studies like this allow epidemiologists to predict or understand why there is an increase in ehrlichiosis, a pathogen spread by the lone star tick, in this specific environment. From a public health standpoint, these studies help identify trends amongst humans, animals, and the environment, based on tick migration, behavior, and disease transmission [7].

**Table 2 life-13-02048-t002:** General overview of how the One Health approach is applied to tick-borne pathogens.

Aspect	Description	Examples
Interconnectedness	Recognizes the interrelation of human and animal health, with the environment.	Tick-borne diseases can affect both humans and animals, and their transmission is influenced by environmental factors—climate and habitat changes.
Collaboration	Encourages cooperation between various disciplines and sectors, including human health, veterinary medicine, entomology, ecology, and environmental sciences.	Public health agencies, veterinary clinics, entomologists, and researchers collaborate to share data, conduct joint investigations, and develop comprehensive surveillance and control strategies.
Surveillance	Integrates monitoring efforts across humans, animals, and ticks to gather comprehensive data on tick-borne diseases.	Surveillance programs collect and analyze data on tick abundance, infection rates, and disease cases in both humans and animals to understand disease dynamics and identify risk factors.
Prevention and Control	Promotes a holistic approach to prevention and control strategies that address the various components of the disease system.	Vector control measures, such as habitat management and acaricide application, are implemented to reduce tick populations. Public education campaigns raise awareness about tick bite prevention in both humans and animals.
Research	Encourages interdisciplinary research to enhance understanding of tick-borne diseases, including their epidemiology, ecology, and transmission dynamics.	Research studies explore the impact of environmental factors on tick populations, investigate the efficacy of preventive measures, and develop new diagnostic tools or vaccines.
Policy and Regulations	Guides the development of policies and regulations that support One Health principles and facilitate coordinated efforts in tick-borne disease prevention and control.	Governments establish policies that promote collaboration between human and animal health sectors, allocate resources for surveillance and research, and regulate the use of acaricides or vaccines.

## 4. Surveillance Systems

### 4.1. National Notifiable Diseases Surveillance System (NNDSS)

In the USA, the CDC conducts case surveillance through the National Notifiable Diseases Surveillance System (NNDSS) [53,54]. In the case surveillance process, about 3000 health departments gather and use data on disease cases to protect their local communities. Surveillance through NNDSS includes around 120 cases of infectious diseases, bioterrorism agents, sexually transmitted diseases, and noninfectious conditions. About 2.7 million infectious disease cases are reported through a network of 3000 public health departments from all 50 states [55]. Figure 2 gives annual reported cases of TBDs from the CDC, in the USA, excluding U.S. Territories and non-U.S. residents, from 2016 to 2020. Data from some reporting areas may be incomplete due to the 2019 coronavirus disease (COVID-19) pandemic or due to post-reconciliation data updates that could not be confirmed or included in the final data set. Lyme disease, followed by Anaplasmosis and spotted fever rickettsiosis, have the highest case counts throughout the years.

### 4.2. TickNET and Other Surveillance Initiatives

TickNET, a public health network, was created in 2007 to foster greater collaboration between state health departments, academic centers, and the Centers for Disease Control and Prevention on surveillance and prevention of TBDs [56]. In 2008, approximately 3.4 million tests were conducted and 288,000 infections of *Borrelia* were found using two tier testing. This network helped to define and identify about 130,000 cases of Lyme disease between 1986 and 2018 [57]. Another study conducted in 2017 screened 7292 deidentified clinical samples of suspected Lyme disease, anaplasmosis, ehrlichiosis, or babesiosis identifying five *Borrelia* species: two causing Lyme borreliosis, *B. burgdorferi* (*n* = 25) and *B. mayonii* (*n* = 9), and three relapsing fever, *borreliae*, *B. hermsii* (*n* = 1), *B. miyamotoi* (*n* = 8), and *Candidatus B. johnsonii* (*n* = 1), a species previously detected only in the bat tick [58].

The CDC also provides data on tick distribution across the USA with maps, which is intended to monitor changes in the distribution and abundance of ticks at county level throughout different states [59]. ArboNET, established in 2000 by the CDC and state health departments following the 1999 emergence of the West Nile Virus (WNV), is the national surveillance system for arboviruses. Alongside tracking human disease cases, ArboNET also compiles information on arboviral infections in categories such as viremic blood donors, non-human mammals, sentinel animals, deceased birds, and mosquitoes. Human surveillance for arboviral disease is largely passive, and relies on the receipt of information from physicians, laboratories, and other reporting sources by state health departments [60].

## 5. Challenges and Limitations

### 5.1. Underreporting and Misdiagnosis

The CDC has defined clinical and serologic criteria for the purpose of standardizing Lyme disease surveillance; these criteria were mostly periodically revised [61,62]. Diagnosing Lyme disease continues to pose difficulties for community physicians working in regions with a high risk of the illness. Not identifying erythema migrans, or in other cases, symptoms resembling a viral infection without a rash, can result in the overlooking or postponement of Lyme disease diagnosis. This, in turn, can lead to ineffective use of antibiotics and the possibility of encountering later stages of the disease [63].On the other hand, serological tests for Lyme disease were excessively utilized within a vast healthcare system, often leading to the misinterpretation of positive outcomes. This resulted in incorrect diagnoses and the widespread misuse of antibiotics. The underreporting of genuine positive cases was balanced by the exaggeration of false-positive cases. As a result, the difference between reported and actual Lyme disease incidence might not be as substantial as previously believed [64].

Another example of misdiagnosing is with RMSF due to the possibility of cross-reactivity with antibodies generated in response to less severe forms of rickettsia infections. Within the genus Rickettsia, there are multiple species, some causing less severe illnesses when compared to RMSF. These species can trigger an immune response, resulting in the production of antibodies like those generated during an infection with Rickettsia rickettsii. As a consequence of this cross-reactivity, serological tests employed for RMSF diagnosis may yield false-positive results [65]. 

### 5.2. Lack of Standardization of Surveillance in Human and Non-Human Hosts

In the past few decades, there have been remarkable advancements in the quality, variety, and accessibility of diagnostic technologies. These advancements have greatly enhanced our comprehension of infectious diseases and our capacity to manage them. However, in the case of TBDs, their relatively recent identification, coupled with constraints in resources and the intricate ecological factors involved, have presented persistent difficulties in diagnosis and surveillance [7].The national Tick-borne Diseases Working Group in their report to congress in 2018 advised allocating resources for research and initiatives focused on understanding tick biology and the ecology of TBDs [66]. This included advocating for comprehensive tick surveillance initiatives, particularly in regions outside the Northeast and Upper Midwest regions. Starting from 2018, the CDC escalated funding to state health departments, even those situated beyond the Northeast and Upper Midwest, to bolster tick surveillance endeavors. Additionally, the CDC issued guidelines pertaining to surveillance, including ticks with significant medical importance and the pathogens they carry [67]. Enhancing the efficiency and cost-effectiveness of such programs is crucial for ensuring their long-term viability.

Applying a One Health perspective to TBDs could yield numerous advantages [68]. However, a significant obstacle lies in the existing separation between human and veterinary medicine, which has resulted in a fragmented landscape of disease surveillance, communication, and control systems. For instance, companion animals and livestock often act as early indicators for TBDs, offering veterinarians the potential to play a pivotal role in notifying public health authorities about outbreaks. Similarly, healthcare providers and veterinarians handling infections should not only focus on treating the affected individual or animal but also input relevant data into a shared database. This cooperative approach to information exchange would not only enhance and expedite treatment decisions, but also enrich the comprehension of TBD pathologies and their geographical distribution [69]. Subsequently, merging this data with ecological surveillance could provide insights to various disciplines such as entomology, epidemiology, and public health, thereby refining control strategies and enhancing regional public awareness [70].

## 6. Innovations and Future Directions

### 6.1. Molecular Surveillance Techniques

Diagnostic technologies have significantly advanced in recent decades, enhancing our understanding and treatment of infectious diseases [71]. However, TBDs continue to pose challenges due to their recent characterization, limited resources, and ecological complexity. For instance, Lyme disease, a well-studied TBD, can go unnoticed without its distinctive erythema migrans rash. Its transient nature and diverse clinical presentation make reliable isolation for culture, examination, or PCR difficult [63]. Serology is the most reliable option [72] but lacks sensitivity within the first three weeks of infection [73] Inaccurate or delayed diagnoses can lead to severe chronic diseases for patients. Discrepancies between infectious disease institutions and patient associations arise from unexplained syndromes linked to tick bites. Various bacterial, viral, and protozoal tick-borne pathogens (TBPs) with diverse pathologies necessitate multiple tests [74,75].Given existing limitations and the growing TBD threat, there’s a consensus on the need for new-generation technologies. In this context, numerous molecular diagnostic techniques, such as next-generation sequencing, metagenomics, and PCR, offer enhanced capabilities for identifying new and emerging pathogens [71]. These methods can simultaneously detect a wide range of targets in a single assay. One promising approach involves the use of barcoded magnetic beads as a platform [76].

### 6.2. Syndromic Surveillance Networks

Syndromic surveillance is a surveillance approach that uses medical data from different sources to monitor disease trends in real time and to detect disease outbreaks [77]. 

Efficient monitoring and surveillance of vector-borne diseases (VBDs) are crucial to preventing outbreaks and responding promptly. Despite international regulations, internal obstacles in countries hinder effective monitoring. A strong centralized network of institutions offers adaptability to address VBD threats. This approach supports health promotion and allows for the integration of environmental and climate monitoring into disease surveillance [78].Using CDC National Syndromic Surveillance Program (NSSP) data from January 2017 to December 2019, researchers studied tick bite visits, considering factors like sex, age, U.S. region, and seasonality [79]. Across this 3-year span, 149,364 tick bite visits were recorded. The highest incidence was in the Northeast (110 per 100,000). Seasonal tick bite visits had two peaks: spring–summer and a smaller one in fall, aligning with the activity of the blacklegged tick, *Ixodes scapularis*. Current surveillance systems do not track tick bites as they are not reportable conditions [79]. Monitoring emergency department tick bite visits through syndromic surveillance offers valuable data to predict when and where people are at risk of TBDs. This can guide public health messages like avoiding tick areas, using repellent, and checking for ticks during high-risk periods [80].

### 6.3. Prevention and Vector Management

Mitigating the impact of TBDs hinges on prioritizing prevention strategies. Essential components of these strategies include adopting personal protective measures, such as wearing appropriate clothing and using tick repellents [81,82]. These measures are informed by surveillance data, which offer valuable insights into the periods and locations of heightened tick activity. Nevertheless, conflicting studies propose that most attempts to reduce human exposure to ticks and their associated diseases through environmental interventions have yielded limited results in curbing the escalating incidence of tick-borne illnesses [83]. On the other hand, despite increasing awareness about TBDs, effective vaccines against most TBDs are not available [84], which is further complicated by global shifts, including increasing human migration [85].

### 6.4. Research Priorities and Policy Implications

Surveillance data serves as a foundation for public education and outreach initiatives [86]. Raising awareness about the risks of TBDs and educating the public on preventive measures is crucial. Surveillance information, such as the geographic spread of tick activity and disease prevalence, informs the development of targeted educational campaigns. By providing accurate, localized information, these campaigns empower individuals to take proactive measures to protect themselves. Furthermore, understanding the prevalence of specific diseases in different regions helps prioritize public health resources for education, outreach, and control efforts [87].

## 7. Conclusions

In summary, the utilization of diagnostic testing and comprehensive surveillance data forms the backbone of effective tick-borne disease (TBD) management. These tools play multifaceted roles, exerting a profound influence across various critical domains of our approach to these intricate illnesses. From shaping strategies for prevention and fostering early detection to influencing the evolution of treatment protocols and driving widespread public education campaigns, the importance of informed decision-making cannot be overstated. 

Diagnostic testing encompasses a wide spectrum of techniques, ranging from cutting-edge molecular methods to serological assays. This arsenal equips healthcare providers to identify the presence of tick-borne pathogens swiftly and accurately within afflicted individuals. Moreover, when aggregated on a broader scale, diagnostic data contributes to a deeper understanding of disease prevalence, distribution, and trends, thus enabling the formulation of targeted preventive strategies. The interplay between diagnosis and prevention becomes a cornerstone of our fight against these diseases.

In parallel, the comprehensive collection and analysis of surveillance data offers a panoramic view of the dynamics of TBDs within communities and ecosystems. This surveillance encompasses vectors, reservoir hosts, human cases, animal cases, and environmental factors, weaving together a rich tapestry of insights that inform risk assessments and resource allocation. The role of surveillance data extends beyond the boundaries of clinical practice, impacting policy decisions and resource distribution by providing a robust evidence base. It aids in forecasting potential outbreaks, anticipating shifts in disease patterns due to changing climatic conditions or urbanization, and directing public health campaigns toward the most vulnerable areas. Through the harnessing of these data-driven resources of diagnostic testing and surveillance data, a united front is forged. This unity empowers stakeholders at every level to effectively combat the persistent and evolving threats posed by tick-borne illnesses, fostering a safer and healthier future.

Reported cases of tick-borne diseases (TBDs) in the USA, excluding U.S. Territories and non-U.S. residents from 2016 to 2020. Data from some reporting areas may be incomplete due to the 2019 coronavirus disease (COVID-19) pandemic or due to post-reconciliation data updates that could not be confirmed or included in the final data set.

## Figures and Tables

**Figure 1 life-13-02048-f001:**
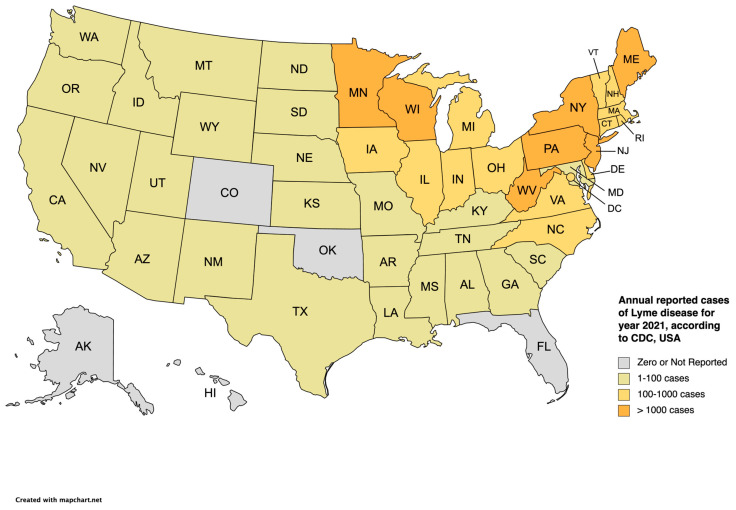
Annual reported cases of Lyme disease according to the CDC in the United States [14] for the year 2021.

**Figure 2 life-13-02048-f002:**
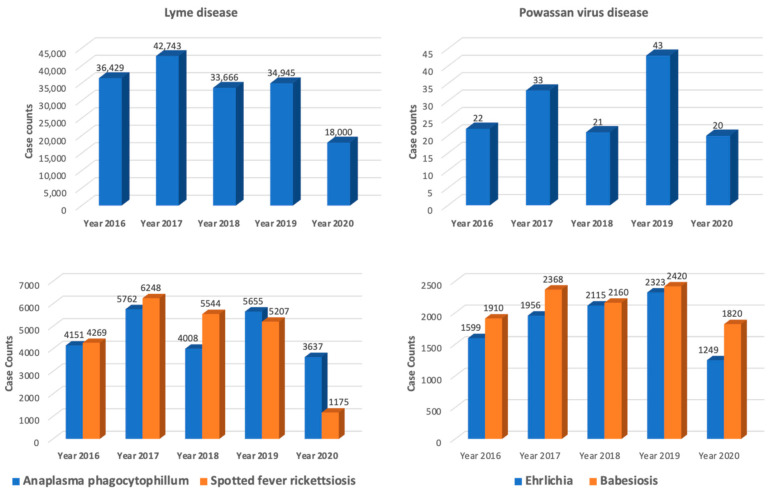
Annual reported cases of tick-borne disease (TBD) according to the CDC in the USA, 2016–2020.

## Data Availability

No new data was created.

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
