# Peer review of "From Tick to Test: A Comprehensive Review of Tick-Borne Disease Diagnostics and Surveillance Methods in the United States"

_life, 2023, doi:10.3390/life13102048_

Round 1
Reviewer 1 Report
The article entitled: “From tick to test: a comprehensive review of tick-borne disease diagnostics and surveillance methods in the United States” submitted by Sean Rowan, Nazleen Mohseni, Mariann Chang, Hannah Burger, Mykah Peters and Sheema Mir briefly presents the methods of surveillance and emphasizes the importance of research on the incidence of tick-borne diseases in the USA. The article is written in a reliable and interesting manner, but there are some flaws that need to be improved before this manuscript can be considered for publication.
Primarily, as the title suggests, in the Introduction Authors should mention ticks species occurring in the United States, and highlight species of the highest medical importance – before tick-borne diseases are described. Then, it will make sense when descriptions of diseases include only a brief reference to the name of the tick species that causes a given disease. Now this approach is inconsistent - because in some cases such a description appears and in others it does not (e.g. L93-96 and L103-119). Please, use scientific names of ticks consistently.
Secondly, Authors should pay their attention to carefully italicize all Latin names of ticks species and genera (e.g.:L142, 157, 194, 300 etc.), check all missing or unnecessary spaces (e.g. L125, 396, 420 etc.) and spelling of diseases (e.g. L333). Additionally, I think that sentence in lines 332-337 should be revised. Moreover, I believe that Figure 2 would be more readable if the years were given in chronological order.
In conclusion, I believe that the manuscript proposed by the Authors should be considered for publication after improving all mentioned flaws.
Author Response
Dear Reviewer 1,
Please find attached our revised review article entitled "From Tick to Test: A Comprehensive Review of Tick-Borne Disease Diagnostics and Surveillance Methods in The United States" for consideration for publication after the edits as suggested by the two reviewers.
We would like to thank you for reviewing our manuscript. We value your comments, and your critic that has greatly improved our manuscript. We have included all suggestions and edits from both the reviewers.
For each comment and edit suggested by the Reviewer 2 below, please see our answer/edit in blue.
Thanks
Sheema Mir
Reviewer 1
The article entitled: “From tick to test: a comprehensive review of tick-borne disease diagnostics and surveillance methods in the United States” submitted by Sean Rowan, Nazleen Mohseni, Mariann Chang, Hannah Burger, Mykah Peters and Sheema Mir briefly presents the methods of surveillance and emphasizes the importance of research on the incidence of tick-borne diseases in the USA. The article is written in a reliable and interesting manner, but there are some flaws that need to be improved before this manuscript can be considered for publication.
Primarily, as the title suggests, in the Introduction Authors should mention ticks species occurring in the United States, and highlight species of the highest medical importance – before tick-borne diseases are described. Then, it will make sense when descriptions of diseases include only a brief reference to the name of the tick species that causes a given disease. Now this approach is inconsistent - because in some cases such a description appears and in others it does not (e.g. L93-96 and L103-119). Please, use scientific names of ticks consistently.
Thank you for your comment. We have included all ticks in US in introduction as suggested. We have included more edits as suggested by reviewer 2 as well.
Secondly, Authors should pay their attention to carefully italicize all Latin names of ticks species and genera (e.g.:L142, 157, 194, 300 etc.), check all missing or unnecessary spaces (e.g. L125, 396, 420 etc.) and spelling of diseases (e.g. L333). Additionally, I think that sentence in lines 332-337 should be revised. Moreover, I believe that Figure 2 would be more readable if the years were given in chronological order.
Thank you for your comment. A few spelling errors were detected, and we authors have careful corrected them along proper writing (italicize all species names in the text).Section 3.1 passive surveillance has been revised (lines as suggested). Figure 2 has been put as a chronological order.
In conclusion, I believe that the manuscript proposed by the Authors should be considered for publication after improving all mentioned flaws.
We had a through look on the manuscript and made suggested edits.

Reviewer 2 Report
This is a timely review on tick-borne disease surveillance that highlights many issues in the field and makes some important points about the requirement for a more unified One Health approach and improved molecular tests for TBD surveillance. Overall, this is a well-structured review on TBDs and their surveillance in the US, but notably lacks information on a number of relevant pathogens and an important source of tick surveillance data.
line 39 & 70: italicise Borrelia burgdorferi
line 70: An additional species Borrelia mayonii, causes Lyme disease and has been identified in the Upper Midwest
line 72, 87: italicise Ixodes scapularis
Figure 1: It is interesting that Minnesota has no cases despite being considered a high incidence state. I imagine this was caused by data collection interruption during COVID. I recommend that the map be updated to show 2021 data (the most recent data available from CDC) to give a more accurate picture of prevalence.
line 85: correct to human granulocytic anaplasmosis
line 90-91: not just New York - this applies to most Lyme-endemic states in the NE and Upper Midwest.
line 123: dengue, yellow fever, and WNV are not tick-borne flaviviruses. This should be corrected to "arthropod-borne" or "vector-borne flaviviruses"
line 126: delete (I) - this is not necessary.
line 142: italicise Amblyomma americanum
section 2.5: this section should also include information on E. muris eauclairensis, a pathogen transmitted by I. scapularis in the Upper Midwest.
line 157-159, 189: italicise the ticks' binomial names.
section 2.6: this section should include some discussion of the fact that many tick-borne rickettsiae are present in the US (e.g. Rickettsia montanensis, R. amblyommatis), and cross-react with the RMSF IFA used for diagnosis, giving an inaccurate picture of actual RMSF infection across the country. There are also other important rickettsial diseases that are not mentioned here, at least R. parkeri rickettsiosis (American boutonneuse fever) and Pacific Coast tick fever should be included. Since Figure 2 shows cases of Spotted fever group rickettsioses rather than RMSF, and CDC now reports rickettsioses as a group rather than reporting RMSF alone, authors should perhaps change this section title to "Spotted fever rickettsioses" to provide information on the main rickettsioses present in the US.
section 2.7: suggest changing the title to "Other Tick-borne Diseases" as diseases beyond CTF and tularaemia are discussed here.
This section should also include Heartland virus and Bourbon virus, two emerging tick-borne viruses.
Table 1: not mentioned in the text. Passive surveillance description should include submission of ticks - "Relies on reports from healthcare providers, laboratories, or the public regarding ticks or diagnosed cases of tick-borne diseases."
Table 2: not mentioned in the text.
line 319: change "of" to "including"
line 323: text and Figure legend are different, and need to be corrected. text says "excluding US territories and non-US residents", legend says "Reported cases of tick-borne diseases in the USA, U.S. Territories, and among Non-U.S. Residents"
section 4.2 should also mention ArboNET, which as well as collecting data on viruses also maps ticks and TBPs at a county level across the US (see https://www.cdc.gov/ticks/surveillance/TickSurveillanceData.html)
section 5.1: could also include the case of RMSF misdiagnosis due to cross-reactivity of less pathogenic rickettsia, as mentioned above.
line 357: TBDs should have been defined in the first line of the manuscript (line 30), not here. Please check throughout as there are multiple places where "tick-borne diseases (TBDs)" is written. Similaly for vector-borne disease and VBD.
line 435: delete the PMCID.
line 459: human and animal cases
Figure 2: are these the most up-to-date data? Also might need to mention in the figure legend that data reporting might have been interrupted due to covid19 - the 2020 disease data look very low for some TBDs.
English language is fine, only a few spelling errors were detected (below), and authors need to be careful to italicise all species names in the text.
line 330: spelling Borrelia
line 333: spelling anaplasmosis, ehrlichiosis
Author Response
Dear Reviewer 2,
Please find attached our revised review article entitled "From Tick to Test: A Comprehensive Review of Tick-Borne Disease Diagnostics and Surveillance Methods in The United States" for consideration for publication after the edits as suggested by the two reviewers.
We would like to thank you for reviewing our manuscript. We value your comments, and your critic that has greatly improved our manuscript. We have included all suggestions and edits from both the reviewers.
For each comment and edit suggested by the Reviewer 2 below, please see our answer/edit in blue.
Thanks
Sheema Mir
Reviewer 2
This is a timely review on tick-borne disease surveillance that highlights many issues in the field and makes some important points about the requirement for a more unified One Health approach and improved molecular tests for TBD surveillance. Overall, this is a well-structured review on TBDs and their surveillance in the US, but notably lacks information on several relevant pathogens and an important source of tick surveillance data.
Thank you for your comment, we have included all edits suggested below.
line 39 & 70: italicise Borrelia burgdorferi
Edited as suggested.
line 70: An additional species Borrelia mayonii, causes Lyme disease and has been identified in the Upper Midwest
Information was added with reference.
line 72, 87: italicise Ixodes scapularis
Edited as suggested.
Figure 1: It is interesting that Minnesota has no cases despite being considered a high incidence state. I imagine this was caused by data collection interruption during COVID. I recommend that the map be updated to show 2021 data (the most recent data available from CDC) to give a more accurate picture of prevalence.
At the time of drafting this article (September 2023), 2021 was not available on CDC but this past month CDC has updated the data from 2021. We have updated the figure with new data available.
line 85: correct to human granulocytic anaplasmosis
Edited as suggested.
line 90-91: not just New York - this applies to most Lyme-endemic states in the NE and Upper Midwest.
Edited as suggested with reference
line 123: dengue, yellow fever, and WNV are not tick-borne flaviviruses. This should be corrected to "arthropod-borne" or "vector-borne flaviviruses"
Edited as suggested.
line 126: delete (I) - this is not necessary.
Edited as suggested.
line 142: italicise Amblyomma Americanum
Edited as suggested.
section 2.5: this section should also include information on E. muris eauclairensis, a pathogen transmitted by I. scapularis in the Upper Midwest.
Added with reference
line 157-159, 189: italicise the ticks' binomial names.
Edited as suggested.
section 2.6: this section should include some discussion of the fact that many tick-borne rickettsiae are present in the US (e.g. Rickettsia montanensis, R. amblyommatis), and cross-react with the RMSF IFA used for diagnosis, giving an inaccurate picture of actual RMSF infection across the country.
Added this information in misdiagnosis section so don’t want to repeat here.
There are also other important rickettsial diseases that are not mentioned here, at least R. parkeri rickettsiosis (American boutonneuse fever) and Pacific Coast tick fever should be included. Since Figure 2 shows cases of Spotted fever group rickettsioses rather than RMSF, and CDC now reports rickettsioses as a group rather than reporting RMSF alone, authors should perhaps change this section title to "Spotted fever rickettsioses" to provide information on the main rickettsioses present in the US.
Title section changes and two other rickettsial diseases added.
section 2.7: suggest changing the title to "Other Tick-borne Diseases" as diseases beyond CTF and tularaemia are discussed here.
This section should also include Heartland virus and Bourbon virus, two emerging tick-borne viruses.
Edited as suggested. Also included two viruses with references
Table 1: not mentioned in the text. Passive surveillance description should include submission of ticks - "Relies on reports from healthcare providers, laboratories, or the public regarding ticks or diagnosed cases of tick-borne diseases.
This information is already added in the table – We have added “(Table1)” to section 3 title.
Table 2: not mentioned in the text.
Added “(Table2)” to section 3.5 title
line 319: change "of" to "including"
Edited as suggested.
line 323: text and Figure legend are different and need to be corrected. text says "excluding US territories and non-US residents", legend says "Reported cases of tick-borne diseases in the USA, U.S. Territories, and among Non-U.S. Residents"–
Edited as suggested.Changed figure legend to “Reported cases of tick-borne diseases (TBDs) in the USA, excluding U.S. Territories and Non-U.S. Residents from 2016 to 2020”. Also changed all years to “2016-2020”
section 4.2 should also mention ArboNET, which as well as collecting data on viruses also maps ticks and TBPs at a county level across the US (see https://www.cdc.gov/ticks/surveillance/TickSurveillanceData.html)
Edited as suggested. Included ArboNET with reference.
section 5.1: could also include the case of RMSF misdiagnosis due to cross-reactivity of less pathogenic rickettsia, as mentioned above.
Edited as suggested.
line 357: TBDs should have been defined in the first line of the manuscript (line 30), not here. Please check throughout as there are multiple places where "tick-borne diseases (TBDs)" is written. Similaly for vector-borne disease and VBD.
Tick-borne disease defined in Abstract line 8, Introduction line 27 and Conclusion line 355, all others have been replaced with “TBDs”. Vector-borne disease defined in section 6.2 line 325, others have been replaced with “VBDs”. Also added “TBD” to keywords.
line 435: delete the PMCID.
Edited as suggested.
line 459: human and animal cases
Edited as suggested.
Figure 2: are these the most up-to-date data? Also might need to mention in the figure legend that data reporting might have been interrupted due to covid19 - the 2020 disease data look very low for some TBDs.
Yes, 2020 is the most up to date data. Added the COVID19 disclaimer to section 4.1 and figure legend.
English language is fine, only a few spelling errors were detected (below), and authors need to be careful to italicise all species names in the text.
Edited as suggested.
Checked grammar and italicized all species name.
line 330: spelling Borrelia
Edited as suggested.
line 333: spelling anaplasmosis, ehrlichiosis
Edited as suggested.

Round 2
Reviewer 2 Report
The authors have improved their nice review with the changes and updates and I recommend that the paper can be published after making the following minor corrections:
lines 36-41; italicise scientific names of ticks
line 40: not sure that D. albipictus is an important vector of any diseases - this species can be deleted from the list
line 61: correct year to 2021 to match the Figure
line 161: Rickettsia philipii and Rickettsia 364D are the same species. Rickettsia philipii is a more recent name given to the species previously known as 364D.
Table 1: Passive Surveillance description - suggest add "ticks" to the description as these often include healthcare professionals, labs, or the public submitting tick samples, as well as reports of TBD diagnoses.
Author Response
Dear Reviewer,
Please find attached our revised review article entitled "From Tick to Test: A Comprehensive Review of Tick-Borne Disease Diagnostics and Surveillance Methods in The United States" for consideration for publication after the edits as suggested by the two reviewers.
We would like to thank you for reviewing our manuscript. We have included all suggestions and edits from both the reviewers.
For each comment and edit suggested by the Reviewer 2 below, please see our answer/edit in blue.
Thanks
Sheema Mir
The authors have improved their nice review with the changes and updates and I recommend that the paper can be published after making the following minor corrections:
lines 36-41; italicise scientific names of ticks
Corrected as suggested
line 40: not sure that D. albipictus is an important vector of any diseases - this species can be deleted from the list
Deleted as suggested
line 61: correct year to 2021 to match the Figure
Corrected as suggested
line 161: Rickettsia philipii and Rickettsia 364D are the same species. Rickettsia philipii is a more recent name given to the species previously known as 364D.
Corrected as suggested
Table 1: Passive Surveillance description - suggest add "ticks" to the description as these often include healthcare professionals, labs, or the public submitting tick samples, as well as reports of TBD diagnoses.
Added as suggested
